# Investigation of Photonic-Crystal-Structured p-GaN Nanorods Fabricated by Polystyrene Nanosphere Lithography Method to Improve the Light Extraction Efficiency of InGaN/GaN Green Light-Emitting Diodes

**DOI:** 10.3390/ma14092200

**Published:** 2021-04-25

**Authors:** Po-Hsun Lei, Po-Chun Yang, Po-Chun Huang

**Affiliations:** Institute of Electro-Optical and Materials Science, National Formosa University, 64 Wen-Hwa Rd, Hu-Wei, Yun-Lin 632, Taiwan; 10676124@gm.nfu.edu.tw (P.-C.Y.); d917205@oz.nthu.edu.tw (P.-C.H.)

**Keywords:** photonic crystal (PC), spin coating method, InGaN/GaN green light-emitting diodes (LEDs), light extraction efficiency (LEE)

## Abstract

We fabricated the photonic-crystal-structured p-GaN (PC-structured p-GaN) nanorods using the modified polystyrene nanosphere (PS NS) lithography method for InGaN/GaN green light-emitting diodes (LEDs) to enhance the light extraction efficiency (LEE). A modified PS NS lithography method including two-times spin-coating processes and the post-spin-coating heating treatment was used to obtain a self-assembly close-packed PS NS array of monolayer as a mask and then a partially dry etching process was applied to PS NS, SiO_2_, and p-GaN to form PC-structured p-GaN nanorods on the InGaN/GaN green LEDs. The light output intensity and LEE of InGaN/GaN green LEDs with the PC-structured p-GaN nanorods depend on the period, diameter, and height of PC-structured p-GaN nanorods. RSoft FullWAVE software based on the three-dimension finite-difference time-domain (FDTD) algorithm was used to calculate the LEE of InGaN/GaN green LEDs with PC-structured p-GaN nanorods of the varied period, diameter, and height. The optimal period, diameter, and height of PC-structured p-GaN nanorods are 150, 350, and 110 nm. The InGaN/GaN green LEDs with optimal PC-structured p-GaN nanorods exhibit an enhancement of 41% of emission intensity under the driving current of 20 mA as compared to conventional LED.

## 1. Introduction

GaN-based light-emitting diodes (LEDs) with a wide and direct bandgap, expected to emit light in visible range by adjusting the composition of indium (In), have attracted much attention and been extensively studied in various applications such as solid-state lighting, visible and underwater communication system, and full-color display [1,2,3]. High external quantum efficiency (EQE), depending on the internal quantum efficiency (IQE) and light extraction efficiency (LEE), is a substantial requirement for GaN-based LEDs in these applications. Many manuscripts have been reported on the design of active region, carrier confinement, and reduction of defects to improve the IQE of GaN-based LEDs. Introduction of the silicon-doped GaN quantum barrier with well-defined thickness in InGaN/GaN active region [4], deposition of short-period superlattices in multiple-quantum-wells (MQWs) [5] and fabrication of lattice-matched InGaN/AlInN/InGaN MQWs [6] enhance the IQE of GaN-based LEDs due to the alleviation of strain in an active region. Electron-blocking layer with high bandgap grown at the interface of MQWs and cladding layer confines the electrons in the active region to increase the probability of radiative recombination in the active region [7]. A current spreading layer with a high transmittance was used for the GaN-based LED as a window layer to expand the injection current into the active region [8]. Tsai et al. reported a novel active region composed of InGaN/GaN MQWs with a GaN capping layer to decrease the appearance of V-defects [9]. Cao et al. also exhibited deposition of multilayer nanoporous GaN as a buffer layer to mitigate the threading dislocations density [10]. These methods have brought blue InGaN/GaN LEDs with remarkable progress in IQE exceeding 80%. However, InGaN/GaN green LEDs with a high In content (25% green and 18% for blue) suffer a lower EQE than blue InGaN/GaN LEDs because of the “green gap” problem caused by the large difference in lattice constant and thermal expansion coefficient between high-In-content InGaN and GaN [11,12]. Several techniques such as deposition of Si-doped graded superlattice or InGaN/GaN superlattice between n-GaN and InGaN/GaN active region [1,13], low-temperature-grown GaN cap layer [14], and AlGaInN/GaN MQWs [15] were used to address this issue.

Although these techniques can advance the IQE of InGaN/GaN green LEDs, however, the LEE of InGaN/GaN green LEDs is low owing to a small critical angle for the generated photons escaped from the p-GaN layer into the air (approximately 23°) caused by the total internal reflection (TIR). As a result, the EQE of InGaN/GaN green LED is still low because a small fraction of photons can be extracted from the device. To increase the LEE of InGaN/GaN green LEDs, several methods based on reducing TIR between p-GaN and air were reported. The common techniques to enhance the LEE of GaN-based LEDs are use of pattern sapphire [16,17,18] and surface roughness [19,20,21]. Designed patterns such as hemisphere, pyramid, or hollow conical to reflect or modify the photons going to substrate direction are fabricated on a sapphire substrate through the etching process. However, the patterned sapphire substrate of GaN-based LEDs requires additional photolithography and dry etching processes before the deposition of the device. The textured window layer or rough surface for GaN-based LEDs accompanied by plasma-based dry etching or chemical wet etching can increase the probability of photons escaped from p-GaN to air. The plasma-based dry etching process will induce radiation damage on the p-GaN surface and bring a significant degradation of electrical properties of the LED and the chemical wet etching process leads to a hard-controlled and nonuniform rough surface depending on the concentration of etching solution and ambient temperature.

Recently, GaN-based LEDs with a periodic photonic crystal (PC) structure to enhance the LEE were widely investigated. To obtain a periodic PC structure, a self-assembly nanosphere (NS) monolayer caused by the balance of van der Waals forces, steric repulsions, and Coulombic repulsions, is used as a temporary pattern layer. There are different strategies to fabricate the self-assembly nanospherical monolayer on a large-area substrate such as dip-coating [22,23], drop-coating [24,25], spin-coating [26,27], sedimentation [28,29], centrifugation [30], electrophoretic deposition [31,32], shear ordering [33], Langmuir-Blodgett [34], self-assembly at the gas/liquid [35] interface, and magnetic self-assembly [36]. Among these methods, spin-coating takes the advantages of simple process, large-area fabrication, low cost, and high throughput to fabricate the self-assembly nanospherical monolayer. However, the spin speed and required time to the set speed have a significant impact on the self-assembly process, leading to the difficulty to obtain a uniform and ordered nanospherical monolayer on a large-area substrate. Using the periodic PC structure on the surface of GaN-based LEDs to improve the escaped probability of photons between p-GaN and air is the most straightforward design. Fu et al. [37] represented SiO_2_-composed nano-honeycomb PC structure by self-assembly polystyrene (PS) pattering to improve LEE and view angle of GaN-based LEDs. Suslik et al. [38] used a multiple exposure process with two-beam interference lithography to define a PC structure on photoresistor and spun liquid polydimethylsiloxane (PDMS) on the patterned photoresistor to form a PC PDMS membrane. The PDMS membrane was then applied to the surface of the LED chip to enhance the electroluminescence (EL) intensity. Hu et al. [39] reported SiO_2_ with a PC structure of square-lattice on the ITO window layer of GaN-based LEDs by UV-nanoimprint lithography. LEDs with the SiO_2_ PC structure show a low turn-on voltage and high light output power. Yin et al. [40] exhibited an improved optical bandwidth and frequency performance of GaN-based LEDs by using nanoholes PC structure with a well-defined period and radius to achieve high-speed operation for visible light communication. Feng et al. [41] presented the enhanced TE- and TM-mode LEE of GaN-based LEDs with cylindrical p-GaN, MQWs, and n-GaN PC structure through the FDTD method. The calculated results also indicate that the TM-mode LEE is higher than TE-mode LEE for GaN-based LEDs with PC structure owing to the light waveguide.

To fabricate the PC-structured p-GaN nanorods on InGaN/GaN green LEDs, we used a self-assembly close-packed polystyrene nanosphere (PS NS) array of monolayer as a mask through the modified PS NS lithography method on a large-area substrate and then obtained the periodic PS NS, SiO_2_, and p-GaN nanorods by partially dry etching process. The modified PS NS lithography method takes the advantage of low cost, easy integration to process, and uniformly large-area fabrication. In addition, LEDs with nano-scaled PC structure show a better performance than those with microscaled PC structure. Uniform nano-scaled and PC-structured p-GaN nanorods can be obtained by using this method. The calculated results indicate that the LEE of InGaN/GaN green LEDs with PC-structured p-GaN nanorods depends on the period, diameter, and height of p-GaN nanorods. The measured electrical and optical characteristics of InGaN/GaN green LEDs with and without the optimal PC-structured p-GaN nanorods are also examined in this study.

## 2. Materials and Methods

Spin-coating PS NS lithography is the simplest approach to fabricate a self-assembly close-packed PS NS array of monolayer on a large-area substrate. However, it is critical to control the experimental parameters such as spin-speed, spin-time, and surface wetting of PS NS suspension on a substrate to obtain a close-packed PS NS array of monolayer on a large-area substrate during the self-assembly process. We investigated the modified PS NS lithography method to achieve this purpose. Figure 1a shows the schematic modified PS NS lithography method incorporated with dry etching process for PC-structured p-GaN nanorods covered indium-tin-oxide (ITO) on InGaN/GaN green LED. The InGaN/GaN green LEDs were composed of a highly Si-doped n-type GaN layer, an InGaN/GaN multiple-quantum-wells (MQWs) active region, and an Mg-doped p-type GaN layer grown on a c-face (0001) sapphire substrate by using a metal-organic chemical vapor deposition system. The PS NS colloidal suspension (Echo Chemical Co., Miaoli, Taiwan) was loaded in a pipette before the modified PS NS lithography process. In the first-time spin-coating process of the modified PS NS lithography method, several liquid droplets composed of PS NS colloidal suspension were dropped on a substrate and then the liquid droplets spread on the substrate after spinning at a high-spin speed for 30 s. Then the substrate was placed on a hot plate at 60 °C for 30 s to obtain a uniform and close-packed PS NS array of monolayer on the large-area substrate. In the second-time spin-coating process of the modified PS NS lithography method, four additional liquid droplets from the pipette were dropped at the four positions of the substrate as shown in Figure 1b and spun at a low-spin speed for 30 s to tailor the uniformity of PS NS array of monolayer at the edge part of the large-area substrate. Table 1 summarizes the parameters of spin-speed and spin-time of the modified PS NS lithography process for PS NS diameters of 100, 200, and 500 nm. A PS NS array with the defined period was formed on an SiO_2_ layer by partially etching close-packed PS NS array of monolayer and then SiO_2_ nanorods with a designed period were obtained on p-GaN layer through partially etching SiO_2_ using a mask of etched PS NS array. All of the etching processes were accomplished by reactive-ion etching (RIE). Finally, PC-structured p-GaN nanorods were obtained by partially removing the p-GaN through inductively coupled plasma (ICP) using a mask of SiO_2_ nanorods. The dry etching processes including RIE and ICP were used to define PC-structured p-GaN nanorods owing to alleviation of distortion for designed PC structure. An ITO was then deposited on the PC-structured p-GaN nanorods as a window layer to spread the injection current.

Finally, the finished wafer was then patterned using the standard photolithographic process to define square mesas as the emitting regions by partially etching the exposed ITO/PC-structured p-GaN nanorods/InGaN/GaN MQWs/n-GaN. A Ti/Pt/Au alloy was used as the ohmic electrode on the ITO and n-GaN contact regions, and the wafer was then alloyed in an N_2_ atmosphere for 5 min at 450 °C. The size of the emission window for the InGaN/GaN LEDs with the PC-structured p-GaN nanorods covered ITO was 300 × 300 μm^2^. The schematic InGaN/GaN LED with the window layer of ITO/PC-structured p-GaN nanorods is displayed in Figure 1c. The ITO layer can serve as the current spreading layer as conventional InGaN/GaN LED. The morphology of the self-assembly PS NS array of monolayer on a glass substrate, SiO_2_ nanorods with PS NS, and PC-structured p-GaN nanorods on InGaN/GaN epi-wafer was measured by using scanning electron microscopy (SEM).

## 3. Results

A well-defined, uniform, and close-packed PS NS array of monolayer on the substrate is necessary to obtain PC-structured p-GaN nanorods. Capillary force occurring between each single PS NS directly determines the arrangement of self-assembly PS NS array of monolayer through spin-coating method [42]. The spin speed, spin time, and droplet volume of PS NS colloidal suspension decide the degree of capillary force between each PS NS, resulting in different arrangement and profile of self-assembly PS NS array [42,43]. A high-order and close-packed PS NS array of monolayer would not be achieved under a high spin speed and low spin time due to the weak adhesive force between nanospheres and substrate [43,44]. Figure 2a shows the plan-view SEM images of the PS NS array with diameters of 100, 200, and 500 nm at the high spin speeds of 3500, 2500, and 1000 rpm under spin time of 30 s and droplet volume of PS NS colloidal suspension of 10 μL/drop. As a large number of PS NSs were spun out of the substrate under a high spin speed (large centrifuge force), as a result, a poor-coverage PS NS array with high-order structure defects was found in Figure 2a [42]. However, decreasing the spin speed will lead to PS NS array of multilayer as shown in Figure 2b, which displays the cross-section SEM images of PS NS array with the diameters of 100, 200, and 500 nm at the spin speeds of 2000, 1300, and 300 rpm under the spin time of 30 s and droplet volume of PS NS colloidal suspension of 10 μL/drop. Low liquid evaporation occurred at low spin speed resulting in a high wetting layer thickness, consequently, a close-packed PS NS array of multilayer was found in Figure 2b owing to the convective particle flux on the substrate for a long time period [44]. Additionally, droplet volume of PS NS colloidal suspension from the pipette is critical in the modified PS NS lithography process. Higher droplet volume of PS NS colloidal suspension from the pipette (13 μL/drop) under the constant spin time of 30 s and spin speeds of 2700, 2000, and 500 rpm for PS NS diameters of 100, 200, and 500 nm (first-time spin-coating process) caused a close-packed PS NS array of multilayer as shown in Figure 2c, which shows the plan-view SEM images of PS NS array with diameters of 100, 200, and 500 nm. The wetting of PS NS colloidal suspension with high droplet volume on the substrate is poor during the spin-coating process; thereby, a PS NS array of multilayer was formed as shown in Figure 2c. In this study, the proper droplet volume of PS NS colloidal suspension from pipette for the modified PS NS lithography process is 10 μL/drop, which enhances the wetting of PS NS colloidal suspension and capillary force between PS NS on the substrate during the spin-coating process.

Figure 3a–c exhibits the plan-view and cross-section SEM images of PS NS array with diameters of 100, 200, and 500 nm at the upper-right, upper-left, lower-right, lower-left, and center parts of a 2 × 2 cm glass substrate using the modified PS NS lithography process and Figure 3d indicates the SEM-measured region on the glass substrate. The optimal heat temperature of the modified PS NS lithography process, which determines the evaporating rate of a solvent of PS NS colloidal suspension on substrate and crystallization of close-packed PS NS array of monolayer after the first-time spin coating process, is around 60 °C [45]. The second-time spin-coating process was used to enhance the rare region of PS NS array that occurred at the four positions between the center and surrounding parts of the substrate as shown in Figure 1c (positions of droplets). Consequently, a uniform and close-packed PS NS array of monolayer with the diameters of 100, 200, and 500 nm was found over the glass substrate in Figure 3a–c due to the proper spin speed, spin time, and droplet volume of PS NS colloidal suspension from the pipette, resulting in a strong adhesive force between the particle and substrate and large capillary force between each PS NS [42,44]. Furthermore, additional spin coating process after second-time spin-coating process leads to a close-packed PS NS array of multilayer as shown in Figure 2c.

The light escape cone of an InGaN/GaN LED is limited by the high refractive index contrast between GaN and air; as a result, it leads to a low LEE. Let **k** be the wave vector of the escape cone; then, [46]**k** = **kN** + **kL**,(1)

Additionally, the in-plane wave vector expresses as follows [46]
**k_L_** = **k_0_** n_GaN_ sinθ_p_(2)
where **k_N_** and **k_L_** are the wave vectors normal to device and in-plane, respectively, the index p labels each mode, **k_0_** = 2π/λ_0_ is the wave vector in vacuum and λ_0_ is the emission wavelength in vacuum and θ_p_ is the angle of propagation for the guided mode. The spatial modulation of the InGaN/GaN LED with PC-structured p-GaN nanorods can be represented by reciprocal vector **G** in reciprocal space and it couples the guided mode to the Bloch mode as a leaky mode out of the PC-structured p-GaN nanorods with satisfying the following diffraction condition [47]
|**k`_L_**| = **|****k_L_** − m**G_0_**| < **k_0_**(3)
where **k`_L_** is a modulated wave vector considering the introduction of PC-structured p-GaN nanorods, m is an integer determining the harmonic orders. Then, the in-plane wave vector is given by [47]
**k`_L_** = **k_L_** + c**k_P_****_C_**(4)
where **k_PC_** is the wave vector of PC-structured p-GaN nanorods, given by [47]
**k_PC_** = (2π/*x_λ_*) **ȃ_x_** + (2π/*y_λ_*) **ȃ_y_**(5)
where *x_λ_* and *y_λ_* are periods in the x and y directions of PC-structured p-GaN nanorods. Changing the periods in the *x* and *y* directions to modulate **k_PC_** can improve the limited light escape cone. However, the diffraction condition in equation (3) depending on the wavelength, lattice constant, and mode propagation angle, and each guided mode propagating at different angles and interacting with PC-structured p-GaN nanorods are all varied. A complex numeric calculation is required to find the optimal period, diameter, and height of PC-structured p-GaN nanorods to achieve a high LEE. We used the RSoft FullWAVE software based on a finite-difference time-domain (FDTD) algorithm to calculate the extracted light intensity of InGaN/GaN green LEDs without and with PC-structured p-GaN nanorods with varied periods in the *x* and *y* directions. After setting the parameters, the distributions of the electric and magnetic field and light output intensity of light-wave emitting from radiation source were computed. The large dielectric constant contrast between p-GaN and air causes a strong TIR; thereby, it reduces the LEE of InGaN/GaN green LEDs. The PC-structured p-GaN nanorods with a designed period, diameter, and height can be used to improve the inherently low LEE of InGaN/GaN green LEDs because of its periodically varied dielectric constant [48]. A photonic bandgap (PBG), which forbid the propagation of light in a range of frequencies, can form in a PC structure for sufficiently large refractive index contrast of PC. Figure 4a shows the PBG band diagram of transverse-electric (TE)- and transverse-magnetic (TM)-like mode for PC-structured p-GaN nanorods with the period, diameter, and height of 150, 350, and 110 nm. The TE-like PBG, which forbids the light propagation in a range of frequencies along the lateral direction, can enhance the light extraction efficiency of LED by redirecting trapped light into radiated modes [49]. Additionally, the period, diameter, and height of the PCs should be to the light wavelength entering PCs to obtain a proper **k_PC_**, implying that the InGaN/GaN green LED with a nano-sized PC structure is better than that with a micro-sized PC structure. Figure 4a–d shows the calculated LEE of InGaN/GaN green LEDs without/with PC-structured p-GaN nanorods with a diameter of 50–450 nm and height of 50, 90, 110, and 150 nm as a function of the period of 50–450 nm. As the period and height of PC-structured p-GaN nanorods increase from 50 to 150 and 50 to 110 nm, the LEE rises gradually because of the formation of coupled modes [50]. However, further increasing the period and height of PC-structured p-GaN nanorods will reduce the LEE possible due to the breaking of coupled modes. The highest LEE occurs at InGaN/GaN green LED with PC-structured p-GaN nanorods of 250 nm period, 200 nm diameter, and 110 nm height in Figure 4c; however, it cannot be performed through PS NS lithography method owing to the size of PS NSs. Based on feasibility and calculated results, the optimal period, diameter, and height of PC-structured p-GaN nanorods are 150, 350, and 110 nm.

Figure 5a,b shows the plan-view and cross-section of SEM images for the dry-etched PS NS/SiO_2_ of 70-nm thickness and Figure 5c displays the cross-section SEM image for the dry-etched PC-structured p-GaN nanorods. SF_6_ with a slow etching rate for PS NS was used to etch close-packed PS NS and SiO_2_ layer to obtain the required patterns. As shown in Figure 5b, the period of each PS NS is almost the same owing to the anisotropic etching process [51] and the period, diameter, and height of periodic SiO_2_ nanorods etched by SF_6_ plasma are about 151, 357, and 70 nm observed in Figure 5a,b. To avoid chemical pollution during removing PS NS, the epi-wafer with PS NS on periodic SiO_2_ nanorods was directly transferred to the ICP chamber for PC-structured p-GaN nanorods etching. After partially etching the p-GaN and removing the PS NS and SiO_2_, PC-structured p-GaN nanorods with a period, diameter, and height of 151, 357, and 112 nm were obtained as can be seen in Figure 5c. These results are close to the calculated optimal PC-structured p-GaN nanorods.

Figure 6a,b plots the forward voltage and light output intensity as a function of injection current (L-I-V) for InGaN/GaN green LEDs without and with varied period and height of PC-structured p-GaN nanorods. The insets of Figure 6a exhibit lighting photographs of InGaN/GaN LEDs without and with PC-structured p-GaN nanorods under the driving current of 20 mA. The turn-on voltage for InGaN/GaN green LEDs with/without PC-structured p-GaN nanorods is around 2.77 V due to the same epitaxial structure. The series resistances for InGaN/GaN LEDs without and with PC-structured p-GaN nanorods of 100, 150, and 200 nm period (fixed height of 110 nm) are 16.7, 20.7, 21, and 20.3 Ω in Figure 6a and for InGaN/GaN LEDs with PC-structured p-GaN nanorods of 90, 110, and 130 nm height (fixed period of 150 nm) are 20.8, 21, and 20 Ω in Figure 6b. With considering the series resistivity, the forward voltage for InGaN/GaN LEDs can be expressed as follows [46]
**V_F_** = **V_D_** + **I_D_R_s_**(6)
where **V_F_**, **V_D_**, **I_D_**, and **R_s_** are forward voltage, diode voltage, forward current, and series resistance. The calculated **V_D_** for InGaN/GaN green LEDs without and with PC-structured p-GaN nanorods is around 3.06 and 3.04 V under driving current of 20 mA. Additionally, the series resistivity for InGaN/GaN green LEDs with PC-structured p-GaN nanorods is higher than that of conventional LED because of the nonuniform distribution of electric field in contact region and surface states. The former is caused by the nano-scale-roughed contact layer and the latter resulted from the bombardment damage during the ICP etching process. A high series resistance will lead to a low light output intensity; however, the InGaN/GaN green LED with the PC-structured p-GaN nanorods show a higher light output intensity than that of conventional LED, which is attributed to improved LEE owing to the PC-structured p-GaN nanorods. The InGaN/GaN green LEDs with the p-GaN nanorods having a fixed height of 110 nm, diameters of 400, 350 and 300 nm, and periods of 100, 150, and 200 nm show the enhanced light output intensity by 19, 41, and 30% as compared to conventional LED in Figure 6a. As the period of PC-structured p-GaN nanorods increases from 100 to 150 nm, the formation of couple mode and better control over the directionality of emission light make the extra extraction of the light modes originally guided within the device, enhancing the light output intensity and LEE of InGaN/GaN green LED. However, the light output intensity decreases with further increasing the period of PC-structured p-GaN nanorods to 200 nm owing to the broken coupled mode. Additionally, the InGaN/GaN green LEDs with the p-GaN nanorods of a constant period and diameters of 151 and 357 nm, and heights of 90, 110, and 130 nm represent the enhanced light output intensity by 19%, 41%, and 34% with comparing to the conventional LED in Figure 6b. The LEE of InGaN/GaN green LED with PC-structured p-GaN nanorods rises with increasing the height of PC-structured p-GaN nanorods from 90 to 110 nm because the PC-structured p-GaN nanorods is close to the active region (MQWs), leading to a strong coupling effect to extract the light emitted from the active region. However, the light output intensity decreases with further increasing the height of PC-structured p-GaN nanorods to 130 nm possibly due to the surface recombination (nonradiative recombination) resulted from bombardment damage during the ICP etching process. The tendency of measured and calculated results is similar but the calculated optic intensity and measured light output intensity are different. These can be attributed to (1) the series resistance including contact resistance and device, which brings about joule heating to increase the junction temperature and turn-on voltage, not considered in the calculated results, and (2) the nonradiative recombination caused by surface state and defects during dry etching process that is not included in the calculated process. Therefore, the calculated optic intensity and LEE is higher than that of measured light output intensity. Figure 6c displays the calculated spectra as a function of the measured angle for InGaN/GaN green LED with the optimal PC-structured p-GaN nanorods and conventional LED under the emitted wavelength at 525 nm. The InGaN/GaN green LED with optimal PC-structured p-GaN nanorods shows the spectrum with higher intensity over the measured angle as compared to that of conventional LED because the PC-structured p-GaN nanorods satisfy the condition of guide mode diffracted from the active region into the air. The PC-structured p-GaN nanorods will suppress the lateral guided mode in the device due to the modified ck_PC_ shown in Equation (4); thereby, InGaN/GaN green LED with PC-structured p-GaN nanorods exhibits a lower calculated intensity as compared to the conventional LED at angles of 90° and −90°. Figure 6d shows the light output intensities under the injection current of 20 mA for the selected chips at different positions of InGaN/GaN wafers with optimal PC-structured p-GaN nanorods. As the period and height of the PC-structured p-GaN nanorods on the InGaN/GaN wafers were relatively uniform, the device to device standard deviation of enhancement of emission intensity was about 0.026, and the variations of light output intensity were approximately 0.54 under the same driving current. In summary, the calculated and measured results indicate that the light output intensity of InGaN/GaN green LED with the optimal PC-structured p-GaN nanorods is higher than that of conventional InGaN/GaN green LED owing to the rising LEE that resulted from the extracted light guided within the device through the coupling effect of PC-structured p-GaN nanorods.

Figure 7a,b exhibits the electroluminescence spectra as a function of wavelength for InGaN/GaN green LEDs without/with PC-structured p-GaN nanorods of fixed height of 110 nm, diameters of 400, 350, and 300 nm, and periods of 100, 150, and 200 nm and for InGaN/GaN green LEDs without/with PC-structured p-GaN nanorods of constant period and diameter of 151 and 357 nm, and heights of 90, 110, and 130 nm under the driving current of 20 mA. The full width at half maximum (FWHM) of the emission spectrum was shown in the inset of Figure 7. The peaks of emitting spectra in Figure 7 are located at 525.4–525.7 nm and these approached peak positions indicating that the adoption of a PC structure in an LED device cannot lead to an apparent shift of emitting spectrum [47]. Additionally, the light output intensity and FWHM for the InGaN/GaN LEDs with the optimal periodic and height of PC-structured p-GaN nanorods were stronger and narrower than those of the conventional InGaN/GaN LEDs owing to the formation of coupled modes [50]. To verify the formation of coupling effect of guided mode in LED, we calculated the optic intensity of InGaN/GaN green LEDs with optimal PC-structured p-GaN nanorods under the different emission wavelengths. Figure 7c shows the calculated optic intensity as a function of measured angle for varied wavelengths through the optimal PC-structured p-GaN nanorods and the 525 nm wavelength represents the strongest light output intensity as compared to other wavelengths. The guiding light emitted from the InGaN/GaN active region underwent TIR and the phase could not match the radiation modes when the amplitude of the in-plane wave vector in the semiconductor was higher than that in air. The InGaN/GaN green LEDs with the optimal PC-structured p-GaN nanorods could adjust the amplitude of the in-plane wave vector in the semiconductor to less than that in air, and therefore, the light was extracted from the semiconductor by the PC-structured p-GaN nanorods because the phase of the guided modes matched the radiation modes, resulting in a high light output intensity and narrow emission spectrum.

## 4. Conclusions

The LEE of InGaN/GaN green LED was improved by PC-structured p-GaN nanorods because of the alleviation of the TIR between p-GaN and air and enhancement of the guided light in the device. The modified PS NS lithography method and followed dry etching process were used to obtain a large-area and uniform PC-structured p-GaN nanorods. The calculated and experimental results indicate that InGaN/GaN green LEDs with PC-structured p-GaN nanorods show a higher LEE and light output intensity than that of conventional LED. InGaN/GaN LED with the optimal PC-structured p-GaN nanorods constructed of the period, diameter, and height of 150, 350, and 110 nm achieves a 41% enhancement of LEE. The device to device standard deviation of measured enhancement of emission intensity was about 0.026 under the same driving current, indicating relatively uniform and reliable PC-structured p-GaN nanorods on the InGaN/GaN wafer.

## Figures and Tables

**Figure 1 materials-14-02200-f001:**
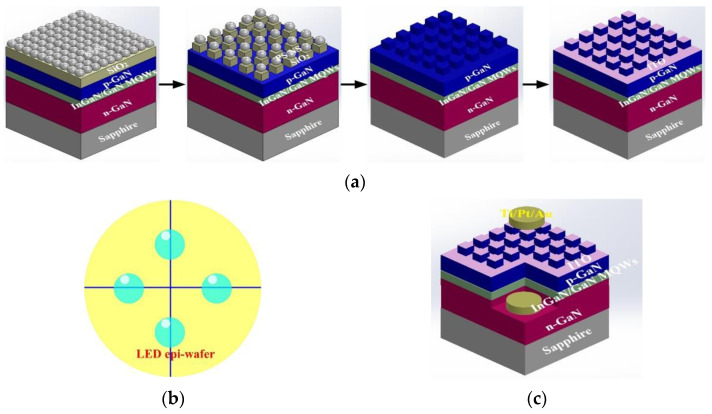
Schematic illustrations of (**a**) process of the PC-structured p-GaN nanorods, (**b**) positions on the substrate for the second-time spin-coating process, and (**c**) the finished device.

**Figure 2 materials-14-02200-f002:**
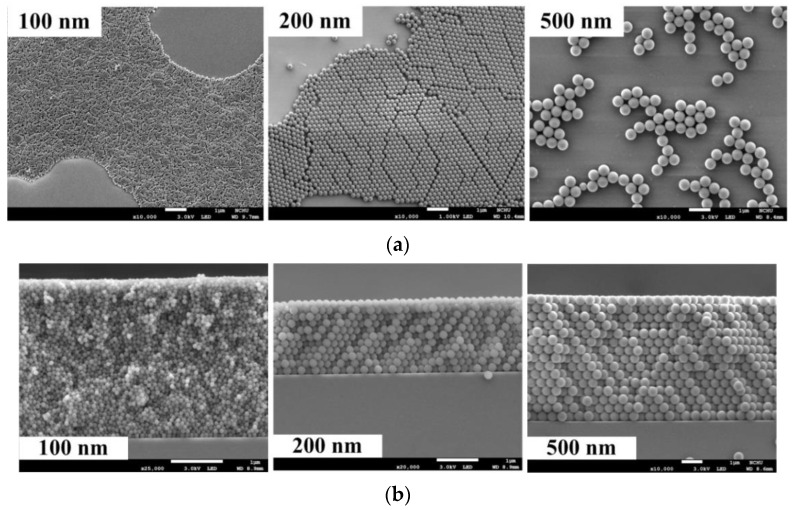
SEM images of PS NS array with diameters of 100, 200 and 500 nm under (**a**) the high spin speed, (**b**) the low spin speed, and (**c**) high volume of the droplet of PS NS colloidal suspension from the pipette (13 μL/drop).

**Figure 3 materials-14-02200-f003:**
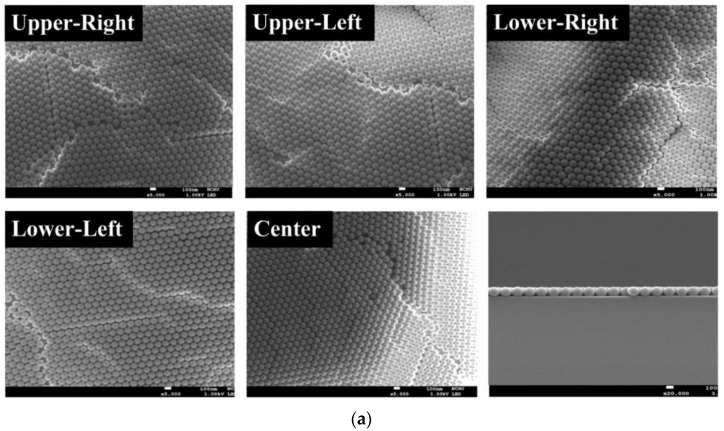
Plan-view and cross-section SEM images of PS NS array with diameters of (**a**) 100, (**b**) 200, and (**c**) 500 nm and (**d**) SEM-measured region on glass substrate for SEM images of (**a**–**c**).

**Figure 4 materials-14-02200-f004:**
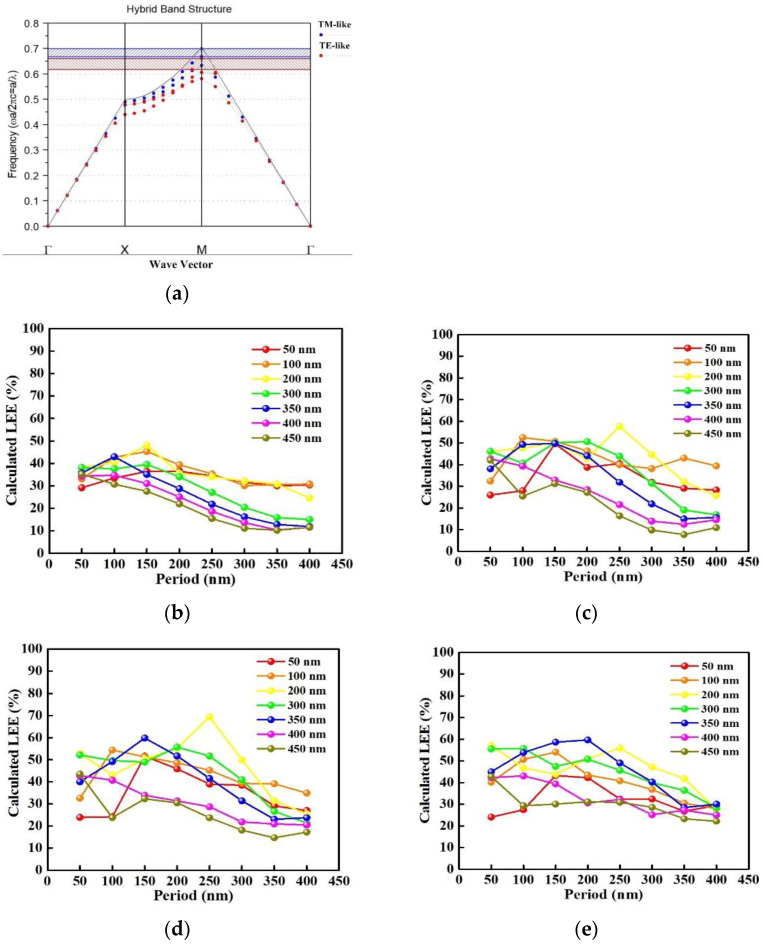
(**a**) The band diagram of PC-structured p-GaN nanorods and the calculated LEE of the PC-structured p-GaN nanorods with the varied period and diameter at a height of (**b**) 50 nm, (**c**) 90 nm, (**d**) 110 nm, and (**e**) 150 nm.

**Figure 5 materials-14-02200-f005:**
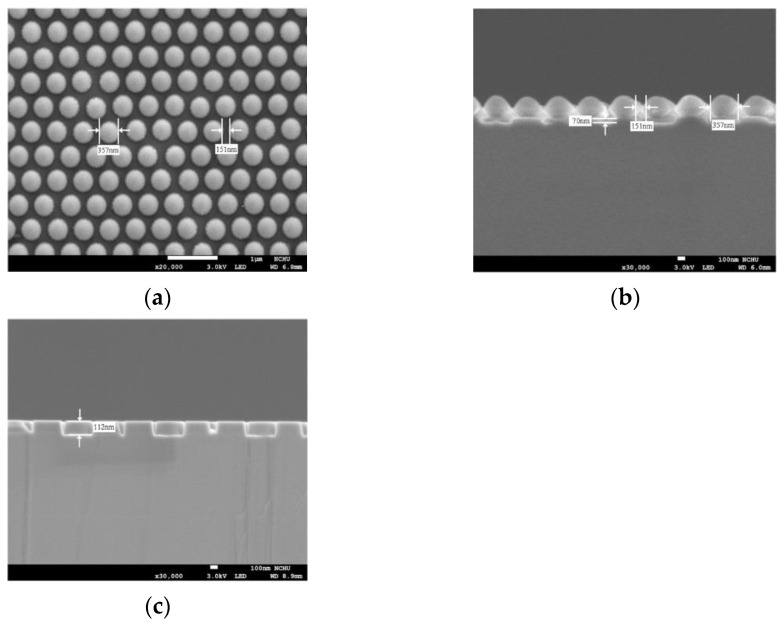
SEM images of (**a**) plan-view and (**b**) cross-section of the dry-etched PS NS/SiO_2_ of 70 nm thickness and (**c**) cross-section of dry-etched 112 nm PC-structured p-GaN nanorods.

**Figure 6 materials-14-02200-f006:**
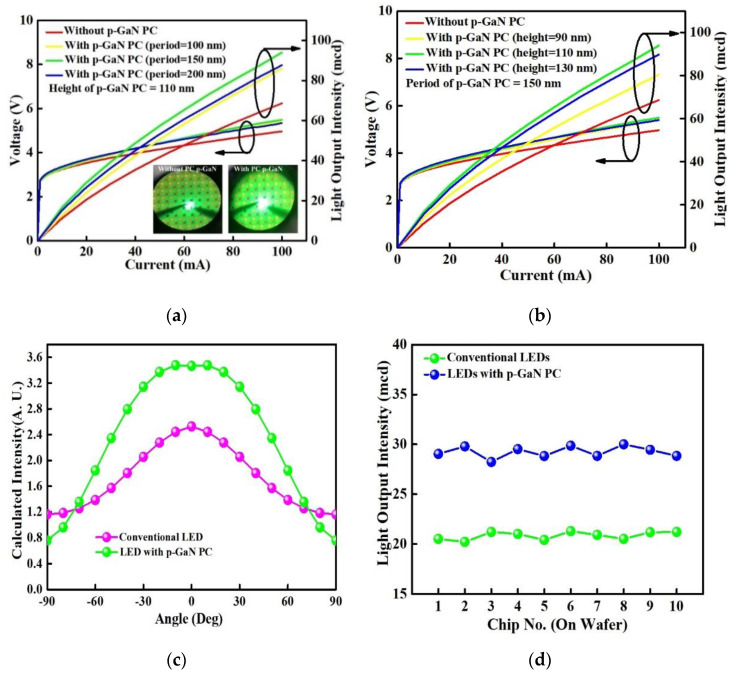
L-I-V measurement for InGaN/GaN green LEDs without and with varied (**a**) period and (**b**) height of the PC-structured p-GaN nanorods. (**c**) The calculated intensity at the measured angle for InGaN/GaN green LEDs without and with the optimal PC-structured p-GaN nanorods at 525 nm, and (**d**) the light output intensity at 20 mA for selected chips on the InGaN/GaN wafer with optimal PC-structured p-GaN nanorods.

**Figure 7 materials-14-02200-f007:**
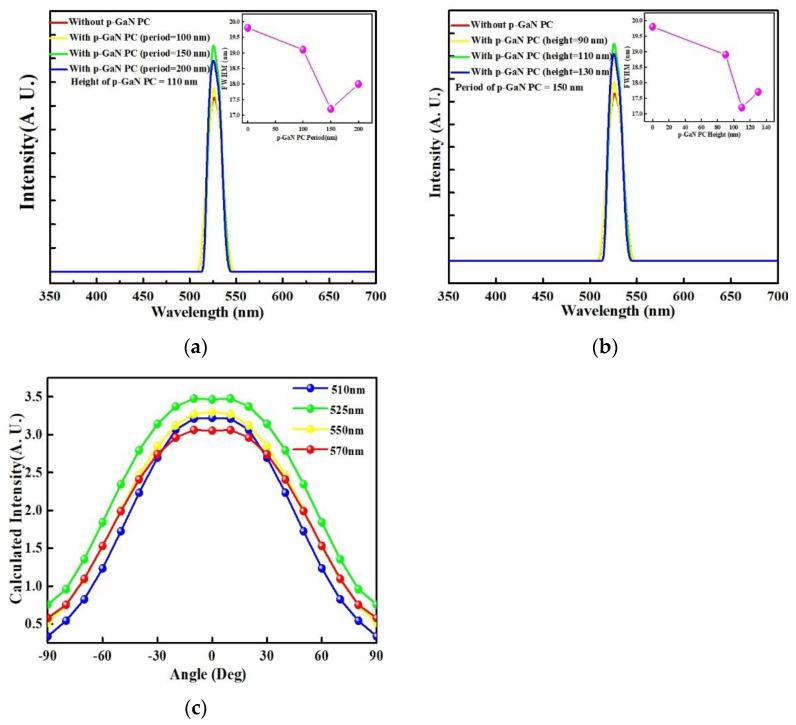
Electroluminescence spectra as a function of wavelength for the InGaN/GaN green LEDs without and with varied (**a**) period and (**b**) height of the PC-structured p-GaN nanorods. (**c**) The calculated intensity of InGaN/GaN green LEDs with the optimal PC-structured p-GaN nanorods at 510, 525, 550, and 570 nm.

**Table 1 materials-14-02200-t001:** The parameters of the modified PS NS lithography process for the varied PS NS diameter.

PS NS Diameter (nm)	First-Time Spin-Coating (Spin Speed/Spin Time)	Second-Time Spin-Coating (Spin Speed/Spin Time)
100	300 rpm/10 s → 2700 rpm/30 s	300 rpm/10 s → 2000 rpm/30 s
200	300 rpm/10 s → 2000 rpm/30 s	300 rpm/10 s → 1300 rpm/30 s
500	500 rpm/30 s	300 rpm/30 s

## Data Availability

The data presented in this study are available on request from the corresponding author.

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
