# Peer review of "Investigation of Photonic-Crystal-Structured p-GaN Nanorods Fabricated by Polystyrene Nanosphere Lithography Method to Improve the Light Extraction Efficiency of InGaN/GaN Green Light-Emitting Diodes"

_materials, 2021, doi:10.3390/ma14092200_

Round 1
Reviewer 1 Report
Authors report their experimental methods and conditions in detail with the extracted results. Their extracted L-I-V curves support that the adopted photonic crystal structures may increase the LEE of InGaN/GaN MQW green LEDs. However, the large portion of supporting theoretical results seems inappropriate, and scientific rigor becomes low. I recommend resubmission after adding the below contents.
1. Please use the correct terminology. For example, guild mode --> guided mode, PS of Equation 4 --> PC.
2. Showing photonic band structures is essential in order to support the physical mechanism of LEE increase in this study. Then, please expand your logic based on the band structure.
3. Fig. 1(c) shows that quite complex photonic crystal slabs are created by Air/ITO/p-GaN/(ITO+p-GaN) layers. In this case, complex photonic band structures are predicted in the slabs. How did you utilize such band structures to increase LEE?
4. Equation (6) is not for the 3D FDTD. In the abstract part, the authors mentioned that 3D FDTD is used for this study. Please correct the equation or the abstract.
5. 3D FDTD can calculate LEE with percentage. Why are you showing the light intensity instead of LEE? The light intensity can not rigorously judge the relative LEE increment among different structures in FDTD. It is because the generated optical power by the soft source can be influenced by PC structures. LEE should be calculated by the (escaped optical power / generated optical power by a dipole).
6. In Figure 5, I-V curves are all different from each other, which means identical epitaxial qualities cannot be guaranteed among different samples. In this case, how to correctly prove the LEE enhancement by the adopted PCs.
Author Response
Reply to reviewers’ comments
In the comments made by the reviewer, some technical and writing comments made from the reviewer for this paper are useful. We thank the constructive comments and made the following revisions. According to reviewer’s comment, we have divided Fig. 2 into Fig. 2 and 3, and Fig. 3, 4, 5, 6, 7 have been rearranged to Fig. 4, 5, 6, 7. The specific comments made by the reviewers and rebuttals are listed as following.
Reviewer 1
1. Please use the correct terminology. For example, guild mode --> guided mode, PS of Equation 4 --> PC
Rebuttal (R1): It is a typo. We thank reviewer’s suggestion and have corrected it in red color.
2. Showing photonic band structures is essential in order to support the physical mechanism of LEE increase in this study. Then, please expand your logic based on the band structure.
R2: We have added the explanation of the improved LEE of LED depending on the photonic bandgap (PBG) (a range of frequency) in page 8 and 9 in red color.
“A photonic bandgap (PBG), which forbid the propagation of light in a range of frequencies, can form in a PC structure for sufficiently large refractive index contrast of PC. There are two approaches to utilize PCs to improve LEE including match of trapped waveguide modes within LED to the bandgap of the PC and refractive index periodicity of a PC to diffract waveguide mode above a certain cut-off frequency into externally propagating modes [49]. The first method can block the propagation of light in lateral directions within the structure, leading to an only emission channel for light to exit; however, it is difficult to achieve because creating a planar structure with a sufficiently large refractive index contrast is impractical. The second method can enhance the diffracted modes of light to escape from the device through a proper kPC. Therefore, we use second method to reduce TIR and then improve the extraction efficiency of LEDs.”
3. Fig. 1(c) shows that quite complex photonic crystal slabs are created by Air/ITO/p-GaN/(ITO+p-GaN) layers. In this case, complex photonic band structures are predicted in the slabs. How did you utilize such band structures to increase LEE?
R3: The LEE of LEDs may be increased by the ITO layer because the ITO layer can serve the functions of current spreading and reducing refractive index contrast between air and p-GaN as conventional InGaN/GaN LED (page 4 in red color). To find the effect of PC-structured p-GaN nanorods, we have deposited an ITO layer on PC-structured p-GaN nanorods as on conventional InGaN/GaN LEDs. Additionally, the ITO layer have been considered in the calculated results in this article
4. Equation (6) is not for the 3D FDTD. In the abstract part, the authors mentioned that 3D FDTD is used for this study. Please correct the equation or the abstract.
R4: We thank reviewer’s comment. Equation (6) is a brief theorem for FDTD algorithm and is not important in this article. We have eliminated equation (6).
5. 3D FDTD can calculate LEE with percentage. Why are you showing the light intensity instead of LEE? The light intensity can not rigorously judge the relative LEE increment among different structures in FDTD. It is because the generated optical power by the soft source can be influenced by PC structures. LEE should be calculated by the (escaped optical power / generated optical power by a dipole).
R5: We thank reviewer’s suggestion and have re-draw Fig. 4 (original Fig. 3) in LEE in page 9.
6. In Figure 5, I-V curves are all different from each other, which means identical epitaxial qualities cannot be guaranteed among different samples. In this case, how to correctly prove the LEE enhancement by the adopted PCs.
R6: The series resistance of InGaN/GaN green LEDs with/without PC-structured p-GaN nanorods is different in I-V curves because of the non-uniform distribution of electric field in contact region and surface states caused by the rough contact layer and damage during ICP process. The high series resistance will lead to a low light output intensity; however, the InGaN/GaN green LED with the PC-structured p-GaN nanorods show a higher light output intensity than that of conventional LED, implying a improved LEE occurred by the PC-structured p-GaN nanorods. We have added the explanation in page 10 in red color.
“The turn-on voltage for InGaN/GaN green LEDs with/without PC-structured p-GaN nanorods is around 2.77 V due to the same device structure. However, the series resistance for InGaN/GaN green LEDs with PC-structured p-GaN nanorods is higher than that of conventional LED because of the non-uniform distribution of electric field in contact region and surface states. The former is caused by the nano-scale-roughed contact layer and the latter is resulted from the bombardment damage during the ICP etching process. A high series resistance will lead to low light output intensity; however, the InGaN/GaN green LED with the PC-structured p-GaN nanorods show a higher light output intensity than that of conventional LED, which is attributed to improved LEE owing to the PC-structured p-GaN nanorods.”

Reviewer 2 Report
Interesting work has been done by the authors in designing the modified PS NS lithography method to enhance the light extraction efficiency of InGaN/GaN LEDs. The dependency of the LEE on the height, period, and diameter has also been discussed fairly. After careful study of the idea, I would like to mention some points and queries that may need to be addressed before publishing the paper.
- You have used spin coating two times in your modified PS NS lithography method. Did you try any other way of creating nanodots? For example, magnetic self-assembly, etc... Is spin coating the best approach for creating the required nanodots? The reason for doing spin coating two times. Is doing third-time inefficient, wasteful, and creating multilayers?
- Is it possible to reduce the amount of PS per drop and increase the no of times spinning is done? Will it create a problem of layering on top of each other?
- What is the control variable of each data in the figures should be explained in an easy-to-understand manner.
- The author claimed that the droplet volume was large, therefore structural defects occurred in figure 2(c). (page 5, lines 189-201) Data to support the claim that the cause of defects is droplet volume, such as when the droplet volume is small and/or when there is no second-time spin-coating, are needed.
- What is the meaning of dividing the position from upper right to center in Figure 2? Do the positions have any special meaning? Is there any possibility of defects in the positions of droplets?
- In Figure 3, 250nm (period), 200nm (diameter), and 110nm (height) was the largest condition with light intensity ~8 (A.U.). But why did you choose 150nm (period), 350nm (diameter), and 110nm (height) as an optimal condition? (line 265, page 9). Additionally, the legends on the right sides of the graphs are not labeled or referenced. (line 267)
- In figure 5(c), A reason is not provided for the lower Intensity of LED with PC-structured p-GaN compared to conventional LED at an angle 90 and -90. Please do explain the reason for such an observation.
- Are the data in Figures 6 and 7 all 350nm diameter? The author presented PS NS arrays of various diameters in figure 2. Are there any results for other diameters?
- In this paper, the authors presented predicted values ​​of various factors for light intensity. In Figures 5 and 6, is the ratio of the light intensity ratio and this predicted value similar?
- Figure 6(d) shows the light intensity for optimal PC-pGaN nanorods. Is there any difference between the unoptimized sample or the value of conventional LED?
- In the last paragraph of page 9, the values are wrongly addressed. Please correct them. (Period, diameter, and height). The respective values at lines 275 and 276 do not match line 280. Especially the respective diameter changes from 70nm to 357nm (just addressing is wrong I think)
- Minor comments:
- Some typos were observed in the paper.
- There is repeating information as they have been used multiple times
Author Response
Reply to reviewers’ comments
In the comments made by the reviewer, some technical and writing comments made from the reviewer for this paper are useful. We thank the constructive comments and made the following revisions. According to reviewer’s comment, we have divided Fig. 2 into Fig. 2 and 3, and Fig. 3, 4, 5, 6, 7 have been rearranged to Fig. 4, 5, 6, 7. The specific comments made by the reviewers and rebuttals are listed as following.
Reviewer 2
1. You have used spin coating two times in your modified PS NS lithography method. Did you try any other way of creating nanodots? For example, magnetic self-assembly, etc... Is spin coating the best approach for creating the required nanodots? The reason for doing spin coating two times. Is doing third-time inefficient, wasteful, and creating multilayers?
R1: Based on low cost and large-area process, we have used dip-drop method to fabricate close-packed PS NS array of a monolayer (Nanoscale research letters, 13, 180 (2018)). However, the process time of dip-drop method is longer than that of spin-coating method. We have added the advantages of spin-coating and modified PS NS lithography method in page 2 in blue color and page 3 in green color.
“Among these methods, spin-coating takes the advantages of simple process, large-area fabrication, low cost, and high throughput”
“The modified PS NS lithography method takes the advantage of low cost, easy integration to process, and uniformly large-area fabrication. In addition, LEDs with nano-scaled PC structure show a better performance than those with micro-scaled PC structure. A uniform nano-scaled and PC-structured p-GaN nanorod can be obtained by using this method.”
2. Is it possible to reduce the amount of PS per drop and increase the no of times spinning is done? Will it create a problem of layering on top of each other?
R2: In our investigation, high amount of PS per droplet will lead to a close-packed PS NS array of a multilayer at first-time spin coating process. Furthermore, low amount of PS per droplet at first-time spin coating process results a poor coverage of PS NS array. The amount of PS per droplet is very critical in spin-coating process. Increasing the no of times spinning will leads to a close-packed PS NS array of multilayer (page 6 in blue color).
“Furthermore, additional spin coating process after second-time spin-coating process leads to a close-packed PS NS array of a multilayer as Fig. 2 (c).”
3. What is the control variable of each data in the figures should be explained in an easy-to-understand manner.
R3: We have added the control variable of each data in Fig. 2 in page 5 in blue color.
“Fig. 2(a) shows the plan-view SEM images of the PS NS array with a diameter of 100, 200, and 500 nm at the high spin speed of 3500, 2500, and 1000 rpm under spin time of 30 seconds and droplet volume of PS NS colloidal suspension of 10μl/drop.”
“However, decreasing the spin speed will lead to PS NS array of multilayer as shown in Fig. 2 (b), which displays the cross-section SEM images of PS NS array with the diameter of 100, 200, and 500 nm at the spin speed of 2000, 1300, and 300 rpm under the spin time of 30 seconds and droplet volume of PS NS colloidal suspension of 10μl/drop.”
4. The author claimed that the droplet volume was large, therefore structural defects occurred in figure 2(c). (page 5, lines 189-201) Data to support the claim that the cause of defects is droplet volume, such as when the droplet volume is small and/or when there is no second-time spin-coating, are needed.
R4: The main problem of high volume of PS NS per droplet is formation of NS PS array of a multilayer during first-time spin-coating process for polystyrene nanosphere lithography method. We thank reviewer’s comment and have added the effect of high volume of PS NS droplet in page 5 in blue color.
“Higher droplet volume of PS NS colloidal suspension from the pipette (13μl/drop) under the constant spin time of 30 seconds and spin speed of 2700, 2000, and 500 rpm for PS NS diameter of 100, 200, and 500 nm (first-time spin-coating process) caused a close-packed PS NS array of multilayer as shown in Fig. 2 (c), which performs the plan-view SEM images of PS NS array with a diameter of 100, 200, and 500 nm. The wetting of PS NS colloidal suspension with high droplet volume on the substrate is poor during the spin-coating process; thereby, a PS NS array of multilayer was formed as shown in Fig. 2 (c).”
5. What is the meaning of dividing the position from upper right to center in Figure 2? Do the positions have any special meaning? Is there any possibility of defects in the positions of droplets?
R5: We have added Fig. 3 (d) to show the region of upper-right, upper-left, lower-right, lower-left, and center on glass substrate.
6. In Figure 3, 250nm (period), 200nm (diameter), and 110nm (height) was the largest condition with light intensity ~8 (A.U.). But why did you choose 150nm (period), 350nm (diameter), and 110nm (height) as an optimal condition? (line 265, page 9). Additionally, the legends on the right sides of the graphs are not labeled or referenced. (line 267)
R6: The PC-structured p-GaN nanorods for maximum LEE can not obtain through PS NS lithography method owing to the size of PS NSs. We have added the description in page 9 in blue color.
“The highest LEE occurs at InGaN/GaN green LED with PC-structured p-GaN nanorods of 250-nm period, 200-nm diameter, and 110-nm height in Fig. 4 (c); however, it can not be obtain through PS NS lithography method owing to the size of PS NSs. Based on feasibility and calculated results, the optimal period, diameter, and height of PC-structured p-GaN nanorods are 150, 350, and 110 nm.”
7. In figure 5(c), A reason is not provided for the lower Intensity of LED with PC-structured p-GaN compared to conventional LED at an angle 90 and -90. Please do explain the reason for such an observation.
R7: We have written the explanation in page 11 in blue color.
“PC-structured p-GaN nanorods will suppress the lateral guided mode in device due to the modified ckPC shown in Eq. 4; thereby, InGaN/GaN green LED with PC-structured p-GaN nanorods exhibits a lower calculated intensity as compared to the conventional LED at an angle 90 and -90.”
8. Are the data in Figures 6 and 7 all 350nm diameter? The author presented PS NS arrays of various diameters in figure 2. Are there any results for other diameters?
R8: We have defined the data in Fig. 6 and 7 in detail.
“The InGaN/GaN green LEDs with the p-GaN nanorods having a fixed height of 110 nm, diameters of 400, 350, and 300 nm, and periods of 100, 150, and 200 nm show the enhanced light output intensity by 19, 41, and 30% as compared to conventional LED in Fig. 6 (a).”
“Additionally, the InGaN/GaN green LEDs with the p-GaN nanorods of a constant period and diameter of 151 nm and 357 nm, and height of 90, 110, and 130 nm represent the enhanced light output intensity by 19, 41, and 34% with comparing to the conventional LED in Fig. 6 (b).”
“Fig. 7 (a) and (b) exhibit the electroluminescence spectra as a function of wavelength for InGaN/GaN green LEDs without/with p-GaN nanorods of fixed height of 110 nm, diameters of 400, 350, and 300 nm, and period of 100, 150, and 200 nm and for InGaN/GaN green LEDs without/with p-GaN nanorods of constant period and diameter of 151 and 357 nm, and height of 90, 110, and 130 nm under the driving current of 20 mA.”
9. In this paper, the authors presented predicted values ​​of various factors for light intensity. In Figures 5 and 6, is the ratio of the light intensity ratio and this predicted value similar?
R9: The tendency of predict and experimental results is similar, but the predicted value is higher than the experimental results. We have added the discussion in page 11 in blue color.
“The tendency of measured and calculated results is similar but the calculated optic intensity and measured light output intensity are different. These can be attributed to (1) the series resistance including contact resistance and device, which brings about joule heating to increase the junction temperature and turn-on voltage, does not consider in calculated results, and (2) dry-etching-induced surface state and defects caused by the nonradiative recombination, do not include in the calculated process. Therefore, the calculated optic intensity and LEE is higher than that of measured light output intensity.”
10. Figure 6(d) shows the light intensity for optimal PC-pGaN nanorods. Is there any difference between the unoptimized sample or the value of conventional LED?
R10: We have re-drawn Fig. 6 (d) and added the light output intensity of conventional InGaN/GaN LEDs in Fig. 6 (d).
11. In the last paragraph of page 9, the values are wrongly addressed. Please correct them. (Period, diameter, and height). The respective values at lines 275 and 276 do not match line 280. Especially the respective diameter changes from 70nm to 357nm (just addressing is wrong I think)
R11: It is a typo. We thanks for reviewer’s suggestion and have corrected it.
“As shown in Fig. 5 (b), the period of each PS NS is almost the same owing to the anisotropic etching process [52] and the period, diameter, and height of periodic SiO2 nanorods etched by SF6 plasma are about 151, 357, and 70 nm observed in Fig. 5 (a) and (b).”
12. Minor comments:
- Some typos were observed in the paper.
- There is repeating information as they have been used multiple times
R12: We thank reviewer’s comment and have modified the typos and reduce the repeating information.

Reviewer 3 Report
Lei et. al. investigate the photonic-crystal-structured p-GaN nanorods fabricated by polystyrene nanosphere lithography method to improve the light extraction efficiency of InGaN/GaN green light-emitting diodes. The research idea in very well conceived and the results are very convincing. The overall flow of the paper is good. I believe it would be a great contribution to MDPI Materials. Hence, I would like to accept paper publication after minor revision of the following comments.
1- Check if the abstract is within the 200 word limit stipulated for MDPI Materials?
2- In the last paragraph of introduction section, briefly explain w-the reason why it was necessary to fabricate this device and explain the novelty of this study for readers' understanding.
3- The introduction and materials and methods sections need to be properly arrange into multiple paragraphs for readers' convenience. Every new information comes under the head of a new paragraph.
4- Figure 2 is too big. Split the figure into two figures as Figure 2 (a-c) as one combines figure and Figure 2(d-f) as another combined figure, Such as figure 3 (a-c). Also, the scalebars and their dimensions need to be magnified in all SEM images in Figure 2 and 4 for clarity.
5- Provide bibliographic reference to all the equations.
Author Response
Reply to reviewers’ comments
In the comments made by the reviewer, some technical and writing comments made from the reviewer for this paper are useful. We thank the constructive comments and made the following revisions. According to reviewer’s comment, we have divided Fig. 2 into Fig. 2 and 3, and Fig. 3, 4, 5, 6, 7 have been rearranged to Fig. 4, 5, 6, 7. The specific comments made by the reviewers and rebuttals are listed as following.
Reviewer 3
1. Check if the abstract is within the 200 word limit stipulated for MDPI Materials?
R1: We thank reviewer’s reminder and have re-written the abstract within 200 words.
2. In the last paragraph of introduction section, briefly explain w-the reason why it was necessary to fabricate this device and explain the novelty of this study for readers' understanding.
R2: We have added the explanation the reason why we used PC-structured p-GaN to improve the LEE of InGaN/GaN green LED in page 3 in green color.
“The modified PS NS lithography method takes the advantage of low cost, easy integration to mass production, and uniformly large-area fabrication. In addition, LEDs with nano-scaled PC structure show a better performance than those with micro-scaled PC structure. A uniform nano-scaled and PC-structured p-GaN nanorod can be obtained by using this method.”
3. The introduction and materials and methods sections need to be properly arrange into multiple paragraphs for readers' convenience. Every new information comes under the head of a new paragraph.
R3: We thank reviewer’s suggestion and have re-arranged these sections.
4. Figure 2 is too big. Split the figure into two figures as Figure 2 (a-c) as one combines figure and Figure 2(d-f) as another combined figure, Such as figure 3 (a-c). Also, the scalebars and their dimensions need to be magnified in all SEM images in Figure 2 and 4 for clarity.
R4: We have divided Fig. 2 into Fig. 2 and 3 and modified the image clearly.
5. Provide bibliographic reference to all the equations.
R5: We have added the bibliographic reference to all equation (ref. 48 and 49).

Round 2
Reviewer 1 Report
- Photonic band structures of your photonic crystal structures are required to prove the existence of some meaningful modes with your suggested structures. Without explanation based on the modes, your sentences are only opinions without direct evidence. If there are no meaning modes with your structure, improved LEE may come from a similar effect like roughed cones.
- If different IV- curves come from the different series resistance, how about comparing the ideality factors of the samples? The identical ideality factor can be a supporting result for the identical epitaxial structures.
Author Response
Reply to reviewers’ comments
In the comments made by the reviewers, some technical comments made from the reviewers for this paper are useful. We thank the constructive comments and made the following revisions.
Reviewer 1
- Photonic band structures of your photonic crystal structures are required to prove the existence of some meaningful modes with your suggested structures. Without explanation based on the modes, your sentences are only opinions without direct evidence. If there are no meaning modes with your structure, improved LEE may come from a similar effect like roughed cones.
Rebuttal (R1): We have added the PBG diagram in Fig. 4 (a) to show the forbidden frequency of TE-like guide mode in page 9.
- If different IV- curves come from the different series resistance, how about comparing the ideality factors of the samples? The identical ideality factor can be a supporting result for the identical epitaxial structures.
R2: We have calculated the series resistance and diode voltage without series resistance from IV curve in Fig. 6 (a) and (b). The series resistance for InGaN/GaN green LED with PC-structured p-GaN nanorods is higher than that of conventional LED. Additionally, the calculated diode voltage without series resistance for InGaN/GaN green LED with and without PC-structured p-GaN nanorods is close. All of the results have been shown in page 10 and 11 in red color.

Reviewer 2 Report
Authors provided proper answers to the reviewers' comments.
Author Response
Reply to reviewers’ comments
In the comments made by the reviewers, some technical comments made from the reviewers for this paper are useful. We thank the constructive comments and made the following revisions.
Reviewer 2
- Authors provided proper answers to the reviewers' comments.
R1: We thank reviewer to accept our answers. We have added a PBG band diagram and calculated the series resistance and diode voltage without series resistance in revised edition. The PBG band diagram indicates that the LEE of InGaN/GaN green LED with PC-structured p-GaN nanorods is enhanced by photonic crystal. The close diode voltage represents similar InGaN/GaN green LED epi-wafer.
